# Physical Activity Is Related to Mood States, Anxiety State and Self-Rated Health in COVID-19 Lockdown

Rafael E. Reigal [1] , José A. Páez-Maldonado [2] , José L. Pastrana-Brincones [3] , Juan P. Morillo-Baro [1] , Antonio Hernández-Mendo [1] and Verónica Morales-Sánchez [1,*]

[1] Faculty of Psychology, University of Malaga, Teatinos Campus, 29071 Malaga, Spain; rafareigal@uma.es (R.E.R.); juanpablo.morillo@gmail.com (J.P.M.-B.); mendo@uma.es (A.H.-M.)

[2] Departamento de Informática y Deporte, Pablo de Olavide University, Utrera Road, 41013 Sevilla, Spain; J.a.paezmaldonado@gmail.com

[3] School of Computer Science and Engineering, University of Malaga, 29071 Malaga, Spain; pastrana@lcc.uma.es

\* Correspondence: vomorales@uma.es

**Abstract:** The main goal of this research is to study the relationships between physical activity, mood states and self-rated health in the Spanish lockdown (March 2020–April 2020) due to the state of alarm caused by COVID-19. The participants were 328 people aged between 19 and 59 years (M = 37.06; SD = 10.82). Females comprised 63.70% of the participants, and 36.30% were male. An associative, comparative and predictive design was used in this research. The International Physical Activity Questionnaire (IPAQ), the Profile of Mood State (POMS), the state anxiety scale of the State-Trait Anxiety Questionnaire (STAI) and the General Health Questionnaire GHQ−12 were applied in order to measure the study variables. Both correlation and linear regression analyses were performed, showing that physical activity is positively related to health perception and mood. Similarly, data have shown that moderate physical practice predicts better health perceptions and positive mood states than vigorous physical activity. Specifically, moderate physical activity is the only variable that predicts the anxiety state ($R = 0.22$; $R^2adjusted = 0.05$; $F = 15.51$; $p < 0.001$). In addition, it has been detected that mood is related to the perception of the state of health. Outcomes suggest that practicing moderate physical activity during these types of situations could amortize its negative effects on psychological health and benefit a more positive mental state. Future studies should consider the employment status of the sample to detect possible differences based on this variable.

**Keywords:** physical activity; moods; self-rated health; state anxiety; COVID-19

## 1. Introduction

In 2020, the SARS-CoV-2 virus (COVID-19) generated an exceptional situation in many countries, causing worldwide distress [1,2]. In Spain, a state of alarm was declared on 14 March 2020 by Royal Decree and was published in the Official State Bulletin. This entailed restrictions on people's mobility and many limitations on the development of activities, both professional and recreational. Non-essential activities were stopped, schools and universities taught classes online and sports competitions were canceled, among other examples [3,4]. Overall, it caused a radical change in the way of living, changing habits and social routines [5–8].

COVID-19 has severely affected people's health [9,10]. Among other consequences, problems such as muscle weakness, respiratory problems, cough, coronary involvement, joint pain, fatigue, loss of smell or taste, cognitive alterations, etc., have been described [11–14]. The damage may be transient, although alterations with uncertain prognoses that may remain over time have also been described [15,16]. This has generated multiple scientific studies, with the aim of determining how harmful they could be to humans [17,18].

Overall, the COVID-19 pandemic has impacted people's well-being and quality of life [19,20]. Even among those not suffering from the disease directly, the level of stress

that people are currently living with is high [21]. The imposed mobility restrictions, the use of masks and disinfectant gels, economic uncertainty, working problems, the way people are in contact with each other, etc., all demand a change in the way of living and a great adaptive capacity. There is some evidence pointing to the impact that the pandemic is having on mental health, increasing the number of anxiety and depression cases [22,23]. Lin, Hu, Alias and Wong [24] have observed that anxiety has increased in the Chinese population due to the pandemic caused by COVID-19.

The pandemic caused by COVID-19 has made many people feel fear and worry because of the uncertainty currently being experienced [25]. Certainly, there is much evidence highlighting how the pandemic has impacted factors affecting people's psychological well-being [26,27]. Distortions in mood states have been described due to COVID-19, which have suggested in all likelihood more severe mental health problems [28,29]. Likewise, it has been observed that social restrictions, getting food and the distress caused by this situation are associated with worse health perceptions by people [30,31]. This is a relevant fact, given the known relationship between self-perception of health and the development of physical or mental illness [32,33].

Promoting the growth of an active lifestyle is an essential goal for health improvements [34]. Specifically, there are many studies that have shown a positive relationship between physical practice and mental health [35,36]. Among others, it has been observed that physical practice has a positive impact on mood [37,38], it decreases anxiety and depression symptoms [39,40] and it improves self-perceived health appraisal [41,42]. This phenomenon has been observed both after a single session of physical activity and after a physical exercise program [43], indicating the broad potential such behaviors have in order to improve people's health and well-being.

It has been observed that during the pandemic, mobility restrictions have generated some changes in people's physical activity behaviors [44]. These changes have caused some people to quit their active behaviors or modify their habits, adapting them to the existing options in such phases of the pandemic [45,46]. Given the benefits of activity on physical and mental health, it would be advisable that people do not stop their physical activity and continue practicing within the given circumstances, adapting them to the protocols in place [47]. Studies such as that by Lesser and Nienhuis [48] have highlighted that the practice of physical activity during those months has been related to a lower level of anxiety and a better perception of mental health. Other studies analyzing the effects of physical activity on mental health levels have highlighted a lower level in anxiety and depression symptoms, as well as a better perception of well-being for those who were more active.

Given the relationship between physical exercise and mental health, as well as to explore whether this issue has occurred during the coronavirus lockdown, the aim of this study is to analyze the relationships between the level of physical activity and mood, anxiety state and health perception in a group of adults in the COVID-19 pandemic.

## 2. Materials and Methods

### 2.1. Design

An associative, comparative and predictive design was carried out in this research in order to analyze the relationships between the studied variables, as well as to explore whether physical activity has predicted mood, anxiety and self-rated health.

### 2.2. Participants

A total of 328 people aged between 19 and 59 years (M = 37.06; SD = 10.82) from the de Andalusia region (Spain) participated in the study. Females comprised 63.70% of the participants, while 36.30% were male. The sampling was non-probabilistic, and was selected from March 2020 to April 2020. Specifically, participants were recruited through snowball sampling using social networks. Participants completed questionnaires using an online survey software (Google Forms). People who had suffered from COVID-19 or were

infected at that moment were excluded from the study because they could be conditioned by several sequelae of the disease. Within the whole sample, when the study was taken, 59.45% did not know anyone who had had the disease, 8.23% had friends who had had the disease, 3.05% had a co-worker who had had it, 12.63% had a family member who had had the disease and 16.63% knew someone who was not a friend, co-worker or family member but had had it.

### 2.3. Measurements and Instruments

(a)  Physical activity in the last seven days. Physical activity was assessed using the short form of the International Physical Activity Questionnaire (IPAQ) [49]. The questionnaire consists of seven items (e.g., *How much time did you usually spend doing vigorous physical activities on one of those days?*), asking about how many days and minutes were spent in the last week doing intense, moderate and low-intensity physical activity, as well as how many hours were spent sitting. In this study, we multiplied the days and minutes taken in the last week by each level of physical activity, obtaining the number of minutes per week in each case. In addition, the number of hours spent sitting was also used.

(b)  Mood. Mood was evaluated using a short, 30-item version of the Profile of Mood States questionnaire (POMS) [50]. This version is made up of 30 adjectives and six factors. In this research, the dimensions of anger (e.g., *Upset*), depression (e.g., *Melancholic*), vigor (e.g., *Full of energy*) and tension (e.g., *Nervous*) were evaluated. Questions have been answered on a Likert scale from 1 (a little) to 5 (a lot). The internal consistency analyses were adequate (Cronbach's alpha), showing values between 0.81 and 0.90.

(c)  State anxiety. State anxiety was evaluated using the state anxiety scale of the State-Trait Anxiety questionnaire (STAI) [51]. This scale consists of 20 items (e.g., *I feel uncomfortable*) scored on a Likert scale from 0 (low anxiety) to 3 (high anxiety). The internal consistency analyses were adequate (Cronbach's alpha), showing a value $\alpha = 0.91$.

(d)  Self-rated health. The 12-item version of the General Health Questionnaire [52,53], was used in order to assess health perception. This questionnaire focuses on the psychological components that identify negative health (e.g., *Feeling unhappy and depressed*). A Likert response scale was used, from 0 (no problems) to 3 (presence of problems). Internal consistency analyses were adequate (Cronbach's alpha), showing a value $\alpha = 0.81$.

### 2.4. Procedure

Data were gathered via online surveys. The evaluation tools were implemented in such a way that they could be carried out via computer, cell phone or tablet. The questionnaires offered a description of the study as well as requests for informed consent for the participants.. The data were gathered between March 2020 and April 2020, when the COVID-19 lockdown at home was effective in Spain.

The estimated time for completing the questionnaires was approximately 45 min. The investigators' contact information was provided just in case respondents had any questions. The ethical principles of the Declaration of Helsinki [54] were respected throughout the research process, and the research was approved by the Ethics Committee of the University of Malaga.

### 2.5. Data Analysis

The data were subjected to descriptive and inferential analyses. The normality of the data (Kolmogorov–Smirnov) and the internal consistency of the scales (Cronbach's alpha) were checked. Pearson's bivariate coefficient was used in order to analyze the correlations between variables. The predictive capacity of weekly physical activity time on the variables of mood, anxiety-state and health perception was evaluated by linear regression analysis (successive steps). SPSS version 23.0 has been used for the statistical processing of the data.

## 3. Results

Table 1 shows the descriptive statistics of the variables under study for the whole sample, as well as by gender and age. Furthermore, the Kolmogorov–Smirnov test has shown that the variables were distributed, fulfilling the assumption of normality. Asymmetry values ranged from −0.19 to 1.26, and kurtosis values ranged from −0.63 to 1.29.

**Table 1.** Descriptive statistics and normality analysis.

| | Total Sample | | Male | | Female | | 19 to 39 Years | | 40 to 59 Years | |
| --- | --- | --- | --- | --- | --- | --- | --- | --- | --- | --- |
| | **M** | **SD** | **M** | **SD** | **M** | **SD** | **M** | **SD** | **M** | **SD** |
| min/week of intense PA | 234.53 | 208.04 | 248.00 | 220.15 | 226.87 | 200.96 | 246.56 | 212.21 | 217.97 | 201.75 |
| min/week of moderate PA | 194.34 | 223.14 | 197.52 | 228.64 | 192.52 | 220.48 | 197.95 | 237.36 | 189.36 | 202.67 |
| min/week of low PA | 168.27 | 161.79 | 177.65 | 169.79 | 162.93 | 157.22 | 169.03 | 165.36 | 167.23 | 157.34 |
| Hours/day sitting | 6.90 | 3.50 | 7.17 | 3.57 | 6.75 | 3.47 | 7.38 | 3.67 | 6.24 | 3.16 |
| Cholera | 2.49 | 0.98 | 2.35 | 1.00 | 2.58 | 0.96 | 2.53 | 0.94 | 2.44 | 1.03 |
| Depression | 2.29 | 0.92 | 2.07 | 0.87 | 2.41 | 0.92 | 2.32 | 0.94 | 2.24 | 0.88 |
| Vigor | 3.21 | 0.81 | 3.34 | 0.77 | 3.13 | 0.82 | 3.14 | 0.79 | 3.31 | 0.82 |
| Tension | 2.62 | 1.05 | 2.35 | 0.94 | 2.77 | 1.08 | 2.65 | 1.05 | 2.57 | 1.05 |
| Anxiety state | 1.03 | 0.63 | 1.03 | 0.60 | 1.03 | 0.65 | 1.02 | 0.63 | 1.04 | 0.63 |
| Self-rated health | 1.07 | 0.49 | 0.96 | 0.43 | 1.13 | 0.51 | 1.08 | 0.48 | 1.05 | 0.51 |

NOTE: M = Mean; min/week = Minutes/week; PA = Physical Activity.

Table 2 shows the Pearson bivariate correlation coefficients for the whole sample. As can be seen, there were significant relationships between the studied variables. The most relevant associations ($p < 0.001$) occurred between vigor and the different parameters of physical activity, between weekly minutes walking with health perception, as well as between weekly minutes of moderate physical activity with anxiety status and health perception. In general terms, weekly minutes of moderate physical activity is the type of physical practice most strongly related to the different parameters of psychological health.

**Table 2.** Correlation analysis (Pearson) (whole sample).

| | Minutes/Week of Intense PA | Minutes/Week of Moderate PA | Minutes/Week of Low PA | Hours/Day Sitting |
| --- | --- | --- | --- | --- |
| Cholera | −0.10 | −0.12 * | −0.13 * | 0.03 |
| Depression | −0.13 * | −0.14 * | −0.13 * | 0.11 * |
| Vigor | 0.22 *** | 0.20 *** | 0.19 *** | −0.21 *** |
| Tension | −0.14 * | −0.15 ** | −0.12 * | 0.05 |
| Anxiety-state | −0.11 * | −0.19 *** | −0.15 ** | 0.04 |
| Self-rated health | −0.17 ** | −0.25 *** | −0.19 *** | 0.16 ** |

NOTE: PA= Physical Activity. * $p < 0.05$; ** $p < 0.01$; *** $p < 0.001$.

Tables 3 and 4 show the Pearson bivariate correlation coefficients by gender and age. There were significant relationships between the studied variables, although the results indicate that men and the younger age group (from 19 to 39 years old) present more robust statistically significant correlations between intense physical activity and the psychological variables analyzed. However, women and the older group (from 40 to 59 years old) show higher statistically significant correlations between moderate physical activity and the psychological variables under study.

Table 5 shows the linear regression models (successive steps) generated. The predictor variables were physical activity (intense, moderate and low) during the last seven days, as well as how long they were sitting. Variables excluded in the various cases are not present due to lack of significance ($p > 0.05$). The data meet the assumptions of linearity in the relationship between predictor variables and criterion, as well as homoscedasticity and normal distribution of the residuals, whose mean value is 0 and standard deviation is near 1 (0.99). In addition, the Durbin–Watson values are satisfactory since they are in a range between 1.53 and 1.91 [55].

**Table 3.** Correlation analysis (Pearson) (by gender).

| | Minutes/Week of Intense PA | | Minutes/Week of Moderate PA | | Minutes/Week of Low PA | | Hours/Day Sitting | |
|---|---|---|---|---|---|---|---|---|
| | **Male** | **Female** | **Male** | **Female** | **Male** | **Female** | **Male** | **Female** |
| Cholera | −0.21 * | −0.05 | −0.24 ** | −0.14 * | −0.08 | −0.03 | 0.12 | −0.03 |
| Depression | −0.21 * | −0.08 | −0.18 * | −0.15 * | −0.12 | −0.08 | 0.25 ** | 0.05 |
| Vigor | 0.20 * | 0.22 ** | 0.14 | 0.22 ** | 0.16 | 0.20 ** | −0.26 ** | −0.17 * |
| Tension | −0.18 * | −0.07 | −0.19 * | −0.21 ** | −0.04 | −0.10 | 0.06 | 0.07 |
| Anxiety-state | −0.13 | −0.17 * | −0.09 | −0.23 ** | −0.13 | −0.09 | 0.13 | −0.01 |
| Self-rated health | −0.25 ** | −0.16* | −0.23 * | −0.25 *** | −0.26 ** | −0.11 | 0.28 ** | 0.10 |

NOTE: PA = Physical Activity. * $p < 0.05$; ** $p < 0.01$; *** $p < 0.001$.

**Table 4.** Correlation analysis (Pearson) (by age).

| | Minutes/Week of Intense PA | | Minutes/Week of Moderate PA | | Minutes/Week of Low PA | | Hours/Day Sitting | |
|---|---|---|---|---|---|---|---|---|
| | **19 to 39 Years** | **40 to 59 Years** | **19 to 39 Years** | **40 to 59 Years** | **19 to 39 Years** | **40 to 59 Years** | **19 to 39 Years** | **40 to 59 Years** |
| Cholera | −0.16 * | −0.03 | −0.07 | −0.18 * | −0.11 | −0.16 | 0.05 | −0.02 |
| Depression | −0.17 * | −0.08 | −0.14 | −0.13 | −0.14 | −0.11 | 0.14 * | 0.05 |
| Vigor | 0.25 ** | 0.20 * | 0.18 * | 0.23 ** | 0.16 * | 0.24 ** | −0.28 *** | −0.06 |
| Tension | −0.18 * | −0.09 | −0.12 | −0.21 * | −0.10 | −0.13 | 0.09 | −0.03 |
| Anxiety-state | −0.01 | −0.25 ** | −0.15 * | −0.25 ** | −0.13 | −0.17 | 0.13 | −0.10 |
| Self-rated health | −0.16 * | −0.19 * | −0.22 ** | −0.29 *** | −0.23 ** | −0.14 | 0.21 ** | 0.07 |

NOTE: PA = Physical Activity. * $p < 0.05$; ** $p < 0.01$; *** $p < 0.001$.

**Table 5.** Linear regression analysis.

| Criteria | M | R | R2 | D-W | Predictors | B | T | T | IVF |
|---|---|---|---|---|---|---|---|---|---|
| Cholera | 1 | 0.17 | 0.03 | 1.53 | (Constant) | | 35.74 *** | | |
| | | | | | Low PA | −0.17 | −2.99 ** | 1.00 | 1.00 |
| Depression | 1 | 0.18 | 0.03 | 1.68 | (Constant) | | 38.01 *** | | |
| | | | | | Moderate PA | −0.18 | −3.34 *** | 1.00 | 1.00 |
| | 2 | 0.23 | 0.05 | 1.68 | (Constant) | | 18.21 *** | | |
| | | | | | Moderate PA | −0.16 | −2.96 ** | 0.98 | 1.02 |
| | | | | | Sitting time | 0.13 | 2.40 * | 0.98 | 1.02 |
| Vigor | 1 | 0.22 | 0.04 | 1.65 | (Constant) | | 45.82 *** | | |
| | | | | | Intense PA | 0.22 | 3.99 *** | 1.00 | 1.00 |
| | 2 | 0.27 | 0.07 | 1.65 | (Constant) | | 28.67 *** | | |
| | | | | | Intense PA | 0.18 | 3.37 *** | 0.96 | 1.04 |
| | | | | | Sitting time | −0.17 | −3.16 ** | 0.96 | 1.04 |
| | 3 | 0.30 | 0.08 | 1.65 | (Constant) | | 27.08 *** | | |
| | | | | | Intense PA | 0.14 | 2.85* | 0.87 | 1.15 |
| | | | | | Sitting time | −0.16 | −2.91** | 0.95 | 1.05 |
| | | | | | Moderate PA | 0.14 | 2.42 * | 0.88 | 1.13 |
| Tension | 1 | 0.19 | 0.03 | 1.58 | (Constant) | | 38.14 *** | | |
| | | | | | Moderate PA | −0.19 | −3.48 *** | 1.00 | 1.00 |
| | 2 | 0.23 | 0.05 | 1.58 | (Constant) | | 34.29 *** | | |
| | | | | | Moderate PA | −0.15 | −2.47 * | 0.87 | 1.15 |
| | | | | | Low PA | −0.13 | −2.24 * | 0.87 | 1.15 |
| Anxiety state | 1 | 0.22 | 0.05 | 2.05 | (Constant) | | 25.13 *** | | |
| | | | | | Moderate PA | −0.22 | −3.94 *** | 1.00 | 1.00 |
| Self-rated health | 1 | 0.24 | 0.06 | 1.73 | (Constant) | | 34.96 *** | | |
| | | | | | Moderate PA | −0.24 | −4.38 *** | 1.00 | 1.00 |
| | 2 | 0.30 | 0.09 | 1.73 | (Constant) | | 15.88 *** | | |
| | | | | | Moderate PA | −0.21 | −3.82 *** | 0.97 | 1.03 |
| | | | | | Sitting time | 0.17 | 3.14 ** | 0.97 | 1.03 |
| | 3 | 0.32 | 0.10 | 1.73 | (Constant) | | 15.38 *** | | |
| | | | | | Moderate PA | −0.17 | −2.95 ** | 0.87 | 1.15 |
| | | | | | Sitting time | 0.15 | 2.79 ** | 0.95 | 1.06 |
| | | | | | Intense PA | −0.13 | −2.21 ** | 0.87 | 1.16 |
| | 4 | 0.34 | 0.11 | 1.73 | (Constant) | | 15.21 *** | | |
| | | | | | Moderate PA | −0.14 | −2.26 * | 0.81 | 1.24 |
| | | | | | Sitting time | 0.14 | 2.55 * | 0.94 | 1.07 |
| | | | | | Intense PA | −0.13 | −2.22 * | 0.86 | 1.16 |
| | | | | | Low PA | −0.12 | −2.12 * | 0.89 | 1.12 |

Note: D-W = Durbin–Watson; T = Tolerance Index; IVF = Variance Inflation Factor. * $p < 0.05$; ** $p < 0.01$; *** $p < 0.001$.

The regression model for anger status showed that this variable was predicted by the practice of low-intensity physical activity ($R = 0.17$; $R^2 adjusted = 0.03$; $F = 8.99$; $p < 0.001$). For the depression status variable, the regression model generated included moderate physical activity and sedentary behavior time ($R = 0.23$; $R^2 adjusted = 0.05$; $F = 8.53$; $p < 0.001$). The

prediction model for vigor status included intense physical activity, sitting time and moderate physical activity ($R = 0.30$; $R^2adjusted = 0.08$; $F = 10.88$; $p < 0.001$). On the other hand, the factors of moderate and low physical activity were predictors of state stress ($R = 0.23$; $R^2adjusted = 0.05$; $F = 12.17$; $p < 0.001$). The model for state anxiety showed one predictor variable, moderate weekly physical activity ($R = 0.22$; $R^2adjusted = 0.05$; $F = 15.51$; $p < 0.001$). For health perception, the linear regression model included heavy, moderate and low physical activity, as well as sitting time ($R = 0.34$; $R^2adjusted = 0.10$; $F = 9.92$; $p < 0.001$).

## 4. Discussion

The aim of this study is to analyze the relationships between the level of physical activity and mood, state anxiety and health perception in a group of adults during the COVID-19 pandemic. Results showed statistically significant relationships between the different parameters of weekly physical activity analyzed and the psychological variables under study.

The data show relationships between the practice of physical activity during the COVID-19 lockdown period and the different psychological parameters analyzed. These results satisfy the objective of the work and show that the practice of physical activity is related to better mental health states, as has also been supported by the scientific literature in recent years [35,36]. Moreover, the results are consistent with the outcomes of other research that have revealed a positive relationship between the practice of physical activity and a better mood, lower anxiety symptoms and better self-perceived health [37–42].

There are studies on physical activity versus mental disorder vulnerability that conclude that daily physical activity decreases the risk of mental illness compared to inactive people [56]. According to recent research, physical activity is considered to be an element that can contribute to the improvement of symptoms of depression and anxiety [57]. In addition, several studies have indicated that physical activity is related to strengthening the immune system, as well as certain parameters of cardiovascular and respiratory system functioning, which have an impact on the psychological perception of health and well-being [57].

According to a recent study, it has been indicated that performing moderate to vigorous aerobic physical activity and muscular strength exercises is associated with a lower likelihood of developing symptoms related to mental disorders [58]. Some authors claim that physical exercise is a useful tool for recovery from disorders related to depression and anxiety [59]. In another recent study, it was observed that the existing relationship between physical activity and mental health is defined as the improvement of mood by increasing blood circulation in the brain area, which influences the hypothalamus–hypophysis–adrenal as well as the physiological response to stress [60]. Finally, another research study showed that patients with severe depression who underwent an aerobic physical activity program experienced significant improvements compared to patients who only received psychotropic treatment [61].

The results reveal slight differences by gender and age, although in general terms, the data are similar. However, the younger age group and men showed a stronger relationship between intense physical activity and the different psychological variables studied. However, statistically significant relationships were more important for women and the older group, in which moderate physical activity was related to psychological variables. This could be because female sports culture has been characterized by lower-intensity practice and is also focused on activities aimed to maintain a physical fitness related to health, while there has been greater use of outdoor spaces and a greater diversification of the activities in males [62]. In addition, younger adults are associated to a greater extent with an intensive, competitive and federated practice, which presents better performance in competitive sports [63]. Regular and non-recreational practice has been associated with older people, who usually practice sports sporadically for well-being and health reasons [64].

This research has been contextualized in a global pandemic caused by COVID-19, specifically in a period when there have been home lockdowns and mobility restrictions,

increasing the interest of these results. In these months, people have suffered high and sustained levels of stress, which can leave an important long-term psychological imprint [65,66]. In fact, these circumstances have increased the predisposition to suffer negative emotional and cognitive responses due to the stress suffered, such as fear, worry or discouragement [25,67]. Moreover, in the first months of the pandemic, ambiguous information has been a common denominator, causing great uncertainty and bewilderment [68] and potentially having consequences of a negative nature on people's well-being [69,70]. Therefore, the results obtained by this research suggest that the practice of physical activity could relieve the negative effects caused by this situation, preserving more adequate levels of mental health.

In a social isolation situation such as the one experienced during quarantine, behaviors such as the practice of physical activity could play a fundamental role in the preservation of health. Even when habits and the type of physical activity performed have to be modified, they can be adapted to be performed at home, thus encouraging benefits in relation to mental health by decreasing anxiety symptoms [71,72]. Psychological distress is greater the longer a person is exposed to a social isolation situation [73]. The observed relationship between mental health and physical activity throughout the COVID-19 pandemic is also conditioned by the impact of physical exercise on self-esteem [74] and also reduces some inflammatory processes buffered by physical activity [75].

This research has several limitations. On the one hand, the cross-sectional design of the present work does not allow for confirmation of whether there are causal relationships between the study variables. The only thing that can be highlighted is the associations found between these variables. However, the existing literature suggests that physical exercise can help to preserve levels of mental health during this period. In addition, even though the residual change score method is comparable to the analysis of covariance when groups are randomly assigned, as shown in Kisbu-Sakarya et al. [76], future research could calculate the residual change score method because it is frequently adopted by researchers to test whether groups differ in the amount of change from pre-test to post-test, estimating the initially predicted post-test scores by regressing the post-test scores on the pre-test scores and ignoring group assignment. On the other hand, the sample is not representative of the whole Spanish population. All the participants were from only one region of Spain, so the results should be interpreted in that context. Likewise, even though we had the information provided by the participants about their physical practice during this period, it was not possible to control exactly what type of exercise they performed. Finally, some personal situations faced by the participants in the study were not considered, such as people who had lost their jobs, people who had to work online or personal healthcare. Therefore, more specific investigations should be carried out to explore possible differences between these populations.

However, the findings obtained suggest that practicing physical activity during isolation due to COVID-19 is associated with mental health status improvement. These results agree with the results found in the literature for non-pandemic times [36,69,74]. Specifically, the results of this research have highlighted the relationships between physical activity and a better mood, lower state anxiety and higher perception of health. Likewise, women and the older group show better psychological health when they practiced moderate physical activity; however, men in the younger age group scored better when they engaged in vigorous physical activity. These findings show the importance of performing physical activity to preserve the state of psychological health, even in critical situations such as those caused by the pandemic due to COVID-19.

**Author Contributions:** Conceptualization, J.A.P.-M., R.E.R., J.P.M.-B., J.L.P.-B., A.H.-M. and V.M.-S.; Methodology, J.A.P.-M., R.E.R., A.H.-M. and V.M.-S.; Software, A.H.-M. and V.M.-S.; Validation, J.A.P.-M., J.P.M.-B., J.L.P.-B., A.H.-M. and V.M.-S.; Formal analysis, R.E.R., J.P.M.-B., J.L.P.-B. and A.H.-M.; Data curation, J.A.P.-M. and J.L.P.-B.; Writing—original draft preparation, J.A.P.-M., R.E.R., J.P.M.-B., J.L.P.-B., A.H.-M. and V.M.-S.; Writing—review and editing, J.A.P.-M., R.E.R., J.P.M.-B., J.L.P.-B., A.H.-M. and V.M.-S.; Visualization, J.A.P.-M., R.E.R., J.P.M.-B., J.L.P.-B., A.H.-M. and V.M.-S.;

Project administration, J.A.P.-M., A.H.-M. and V.M.-S. All authors made substantial contributions to the final manuscript. All authors have read and agreed to the published version of the manuscript.

**Funding:** This research received no external funding.

**Institutional Review Board Statement:** The study was conducted according to the guidelines of the Declaration of Helsinki, and approved by the Ethics Committee of University of Málaga.

**Informed Consent Statement:** Informed consent was obtained from all subjects involved in the study.

**Conflicts of Interest:** The authors declare no conflict of interest.

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
