# Peer review of "Physical Activity Is Related to Mood States, Anxiety State and Self-Rated Health in COVID-19 Lockdown"

_sustainability, doi:10.3390/su13105444_

Round 1

Reviewer 1 Report

Dear authors.

Congratulations on your research. The manuscript that you present can contribute a grain of sand to improve the well-being of people when they are in a situation of mandatory or voluntary lockdown.

Here are some considerations that could help improve your manuscript.

Introduction: it is adequate, and you justify the study very well. You state that there is already a proven relationship between physical exercise and mental health and well-being (many studies have discussed this before). This makes the reader wonder how new this study is and encourages anyone to read on to find the answer. Could it be that during lockdown that relationship changed?

Method:

You must begin by defining the type of design that the research raises and why.

In the participants section, you must provide information on the type of sampling used and on what recruitment strategies you have carried out. Is the sample representative of the Spanish population? Did you consider that they were people who had to go to work? Were healthcare personnel exposed to greater stress ruled out? Although you give some information on exclusion criteria (not being ill), the profile of the people you were looking for the research is not sufficiently clear (you must clarify inclusion / exclusion criteria and justify why). It is also important to clarify the scope (Where?) And the period (When?), Although later you cite the time of data collection, it should be better organized when presenting the method.

Results:

They are well exposed and reported.

Discussion:

It is adequate and sufficient, but it needs a section of limitations, where you provide the main limitations regarding the validity of your study (external and internal) and the difficulty of drawing conclusions from a descriptive and associative / correlational study (non-experimental). Keep in mind that you have not sufficiently justified the type of target population, the recruitment process and the possible strange variables have not been controlled or at least, their control has not been reported.

Reviewer 2 Report

The paper is interesting. I suggest adding examples of items relating to the scales included in the questionnaire. It isn't clear if other variables besides gender, age and information related to the covid were asked (for example how they exercised during the lockdown). I would also adda paragrafh on the possible limitations of this study. I recommend reviewing academic english.  

Reviewer 3 Report

Comments to the Authors

Thank you for the opportunity to review this paper. Overall, it is a well conducted and written study on an important topic. Please see one major and some minor concerns related to this paper.

Title

The title contains the key features of the article. Also, the title is attractive and might spot interest in the reader.

Abstract

The abstract is well written and important information is provided for the reader. However, I suggest authors to report some specific results in the abstract (i.e., add some specific values/numbers). The abstract could also end with a suggestion for future research.

Page 1, line 14: “…of the COVID-19. 328 people…”, I would start this new sentence: “The participants were…”.

Page 1, lines 15-16: “It has been used…” – it seems a bit wordy for me.

Introduction

The authors provide adequate review of the existing literature.

Page 1, line 44: I suggest replacing “caused by” with “of”.

Materials and Methods

Overall, materials and methods are adequately described.

Results

Overall, results of the current research are well explained. However, the current sample of the study has a very wide age range. I would ask authors to report results separately for different age groups and importantly, to compare the results of these different age groups (a major concern). I believe this would strengthen the manuscript. Also, please add the comparisons between the groups of gender.

Discussion

Overall, discussion is well written, but I would suggest authors to also discuss about findings about different age groups and possible gender differences (if there are any).

Page 7, line 242: I agree that the cross-sectional design is one of the limitations of the study. The authors could add suggestion for the future research that residual change score approach could be used in the future research to draw more causal conclusions (e.g., Kalajas-Tilga et al., 2021).

Kalajas-Tilga, H., Hein, V., Koka, A., Tilga, H., Raudsepp, L., Hagger, M. S. (2021). Application of the Trans-Contextual Model to Predict Change in Leisure Time Physical Activity. Psychology & Health. https://doi.org/10.1080/08870446.2020.1869741 

Page 7, lines 248-250: I suggest authors to improve this last conclusion paragraph.

Round 2

Reviewer 3 Report

Authors have sufficiently addressed most of the issues raised by the Reviewer. However, I would like Authors to double-check if they have addressed all the issues raised by the Reviewer.

Author Response

We have checked the answers and there were two questions to be solved. We attach these comments with the answers.

Best regards and thanks.

Abstract

The abstract is well written and important information is provided for the reader. However, I suggest authors to report some specific results in the abstract (i.e., add some specific values/numbers). The abstract could also end with a suggestion for future research.

AUTHORS: It has been added: “Specifically, moderate physical activity is the only variable that predicts the anxiety state (R = .22; R2adjusted = .05; F = 15.51; p < .001)………..Future studies should consider the employment status of the sample to detect possible differences based on this variable”.

Page 7, line 242: I agree that the cross-sectional design is one of the limitations of the study. The authors could add suggestion for the future research that residual change score approach could be used in the future research to draw more causal conclusions (e.g., Kalajas-Tilga et al., 2021).

Kalajas-Tilga, H., Hein, V., Koka, A., Tilga, H., Raudsepp, L., Hagger, M. S. (2021). Application of the Trans-Contextual Model to Predict Change in Leisure Time Physical Activity. Psychology & Health. https://doi.org/10.1080/08870446.2020.1869741 

AUTHORS: This text has been added in limitations paragraph: “In addition, even when the residual change score method is comparable to the analysis of covariance when groups are randomly assigned as shown in Kisbu-Sakarya et al. [76], future research could calculate the residual change score method because is one method frequently adopted by researchers to test whether groups differ in the amount of change from pre-test to post-test, estimating initially the predicted post-test scores by regressing the post-test scores on the pre-test scores, ignoring group assignment.”